# The Swiss Narcolepsy Network (SNaNe)

**Claudio L. A. Bassetti** [1,*], **Ramin Khatami** [1], **Silvia Miano** [2], **Elena Wenz** [1] and **Esther Werth** [3]

1. Neurology Department, Inselspital, University of Bern, 3012 Bern, Switzerland; ramin.khatami@barmelweid.ch (R.K.)
2. Centro del Sonno, Neurocentro della Svizzera Italiana, 6900 Lugano, Switzerland
3. Center for Sleep and Stress Medicine, Bellevue Medical Group, 8001 Zürich-Oerlikon, Switzerland
* Correspondence: claudio.bassetti@insel.ch

**Abstract:** The Swiss Narcolepsy Network (SNaNe) was founded in 2017 as a non-profit organization with the vision of improving the care of patients with narcolepsy, central disorders of hypersomnolence (CDH), and rare sleep disorders. The SNaNe aims at maximizing the speed of diagnosis, minimizing difficulties stemming from the rare nature of these conditions, and providing patients with optimum health care throughout the course of their disease. In addition, the SNaNe promotes education, awareness, and research on CDH and rare sleep disorders. The article reports the current structure, organization, and the following main activities of the SNaNe: (1) the discussion of complex patient cases; (2) the organization of the Swiss Narcolepsy Days; (3) the coordination of multicenter research projects (e.g., SPHYNCS and iSPHYNCS studies); (4) the establishment of an anonymous Swiss registry for CDH patients (SNaNe Data Registry); (5) the collaboration with the national patients' organization (SNAG); and (6) the collaboration with other national and international scientific, professional, and patients' (eNAP) organizations.

**Keywords:** narcolepsy; rare sleep disorders; hypersomnolence (CDH)

## 1. Introduction

In Switzerland, a disease is considered rare if it affects no more than 5 out of 10,000 people. Since there are 7000–8000 rare diseases, with about seven (2–12) percent of the population concerned, the total number of patients affected by a rare disease in our country is estimated to be more than half a million. Narcolepsy is a rare disease with a suggested prevalence of 1/2000 (data in the literature on the prevalence vary somewhat and we refer to a recent review for details [1]). Patients suffering from narcolepsy share a common fate with patients affected by other rare diseases. Indeed, there is little, although increasing (we refer to a recent review for details [1]), research in this field and, consequently, overall expertise and medical care are rather limited.

Nevertheless, the number of rare diseases in Switzerland adds up to a large total, which is why, in 2014, the Federal Council adopted the national concept "Rare Diseases", and the Federal Council approved its implementation plan in 2015. This plan comprises 19 measures and the following four projects: the creation of reference centers, cost coverage, information exchange, and research. In 2016, the Swiss Academy of Medical Sciences (SAMS) published recommendations pertaining to the creation and operation of reference centers for rare diseases in Switzerland [2].

In 2017, a group of narcolepsy experts founded the Swiss Narcolepsy network (SNaNe) in Bern for the following main reasons:

1. Primary central disorders of hypersomnolence (CDH), including narcolepsy, are often diagnosed late (to date, the diagnostic latency for these disorders in Switzerland is still >5 years [1]).
2. There is, overall, little expertise for the management of CDH in Switzerland.

3. The «Burden of disease» of CDH, which often begins early in life, is high (comparable to that of patients with epilepsy, for example) [3].
4. The therapy is neither well established, nor reimbursed for most CDH.
5. The need for supportive and psychosocial therapies for CDH is not met in Switzerland.

## 2. Narcolepsy and Primary Central Disorders of Hypersomnolence

Primary central disorders of hypersomnolence (CDH) are sleep disorders with increased daytime sleepiness that is not caused by circadian sleep–wake rhythm disorders, sleep-disordered breathing, or disorders of nighttime sleep [4]. Primary CDH are differentiated from secondary forms of central diseases, in which sleepiness is caused by an overt brain disorder/damage.

According to the ICSD-3 classification [4], primary CDH encompass classic narcolepsy (with cataplexy, NT1; without cataplexy, NT2) and other entities, such as idiopathic hypersomnia and insufficient sleep syndrome (Table 1).

**Table 1.** Primary central disorders of hypersomnolence (CDH).

| |
|---|
| Narcolepsy with cataplexy (NT1) |
| Narcolepsy without cataplexy (NT2) |
| Idiopathic hypersomnia |
| Kleine–Levin syndrome |
| Insufficient sleep syndrome |

Classic narcolepsy (NT1) is a hypothalamic disorder (Figure 1) with typical onset in adolescence or young adulthood. This disorder leads to the severe impairment of quality of life [3,5]. The leading symptoms of NT1 consist of excessive daytime sleepiness and cataplexy, i.e., a brief bilateral loss of tone (possibly resulting in a fall) triggered by sudden emotions (often accompanied by laughter). NT1 results from a confluence of genetic and environmental factors, leading to an immune-mediated loss of hypocretin/orexin-producing neurons in the hypothalamus [3,6]. The diagnosis of NT1 is primarily clinical but it can be supported by reliable neurophysiological (premature occurrence of REM sleep after falling asleep, the so-called Sleep Onset REM (SOREM)), serum (HLA haplotype DQB1*0602), and cerebrospinal fluid (CSF) biomarkers [5]. The treatment of narcolepsy is in many patients successful due to the introduction, over the past 20 years, of several effective symptomatic therapies for daytime sleepiness and cataplexies, which have recently been approved for children, as well [5,7]. However, no causative treatment of narcolepsy is available and strategies to prevent narcolepsy are missing. Also, cognitive deficits, fatigue, and other disturbances are difficult to treat, reducing the vitality, workability, and, eventually, quality of life of patients.

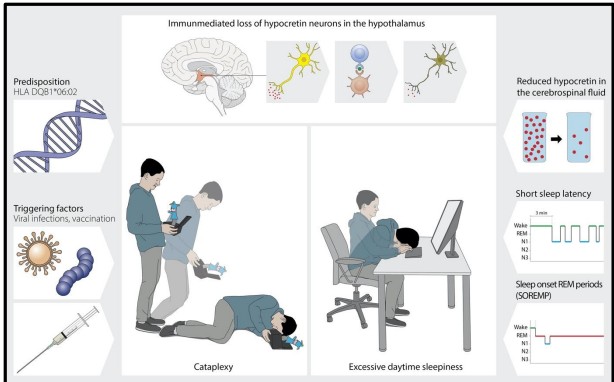

**Figure 1.** Leading symptoms, genetic and non-genetic factors, as well as biomarkers of narcolepsy with cataplexy, an immune-mediated disorder, which is caused by a loss of hypocretin neurons in the hypothalamus. (Figure A. Giger, Bern).

The diagnosis and treatment of the other primary CDH* are difficult and controversial in daily practice due to the lack of reliable biomarkers. Also, the differentiation between primary and secondary CDH (e.g., due to a medical or psychiatric disorder) is often not trivial.

\* For a detailed presentation of all primary CDH, which is not the purpose of this article, we can refer to recent reviews [4,5,8].

### 3. Narcolepsy Medicine and Research in Switzerland: A Long-Standing Tradition

Narcolepsy medicine and research, along with its "borderlands", have a long-standing tradition in Switzerland from the 1950s onwards [9–11]. In 1983, the world's first patient organization, the Swiss Narcolepsy Society (SNaG), was founded in Bern. Two of the last three international narcolepsy congresses were organized by on of the authors (C.L.A.B.) and held in Ticino (Monte Verità, 2004 and 2009).

In 2009, the European Narcolepsy Network was founded by one of the authors (C.L.A.B.) in Zurich, which has since launched several international research projects and conferences under either the co-leadership or involvement of Swiss specialists. The world's largest database on narcolepsy, which was launched in 2010 and currently includes over 2000 patients, is still maintained by Swiss researchers (including one of the authors, R.K., and we refer to recent publications for a detailed presentation of this network [12–14]).

The first European Narcolepsy Day was organized by one of the authors (C.L.A.B) also in Switzerland (Lugano 2012).

### 4. The Swiss Narcolepsy Network (SNaNe): Foundation and Mission

The Swiss Narcolepsy Network (SNaNe) was founded in 2017 in Bern with the aim of improving the health care of patients with NT1, NT2, and CDH (see above). In addition, SNaNe seeks to promote the education, training, and continued education of physicians and other health care professionals concerned, in addition to enhancing the research and awareness of this rare disease group. In order to provide care to local patients, maximize the use of local resources, and promote multicenter collaboration at the national level, the bottom-up formation of a network of centers was given preference, rather than a top-down establishment of a single national center.

In the first general meeting conducted on 3 June 2020, the statutes and board were elected. Subsequently, in addition to individual membership, other forms of membership, including that for institutional/affiliated sleep centers (https://www.snane.ch/affiliate-centers, accessed on 3 June 2020), as well as a financial basis for the consortium, were implemented.

### 5. The Swiss Narcolepsy Network (SNaNe): The First 6 Years

During the first 6 years of its life, SNaNe developed the following main activities, among others:

1. The virtual presentation and discussion on complex patient cases in collaboration with the European Narcolepsy Network (EU-NN; https://www.snane.ch/eventi/snane-case-discussion-webinar-3, accessed on 1 September 2023).
2. The organization of the Swiss Narcolepsy Days (last meeting held in Basel 2023) in collaboration with the SNaG patient organization (https://www.snane.ch/eventi/8th-swiss-narcolepsy-day, accessed on 1 September 2023).
3. The coordination of multicenter research projects, such as the SPHYNCS study, was supported by the SNSF (Swiss National Science Foundation) for the period of 2023-6 [15]. This study (Figure 2) aims to identify new biomarkers for primary central nervous hypersomnia. An international extension of SPHYNCS (so-called iSPHYNCS) was approved by the SNSF for the period of 2023-6. This new study will involve three further international centers, including Bologna (Italy), Witten-Herdecke (Germany), and Leiden (Holland). In addition to the search of new biomarkers, this new

study will investigate both patients' treatment adherence as well as "patient-related outcomes" (PROMs).

4. The establishment of an anonymous Swiss registry for patients with primary CDH (SNaNe Data Registry). The registry is operated by the Clinical Trial Unit (CTU) in Bern according to the latest safety standards and contains information on symptoms, sleep studies, and molecular disease markers. Currently, as of September 2023, more than 170 patients from five different centers have been included in the registry.

5. The exchange and collaboration with the Swiss (SNAG) Narcolepsy society, as well as with the European patient organization (European Narcolepsy alliance for patients, eNAP);

6. The exchange and collaboration at the national and international level with the Swiss Society of Sleep Research, Sleep Medicine, and Chronobiology (SSSSC); Swiss Society of Neurology (SSN); Swiss Society of Neuropediatrics (SSNP); and the European Narcolepsy Network (EU-NN)

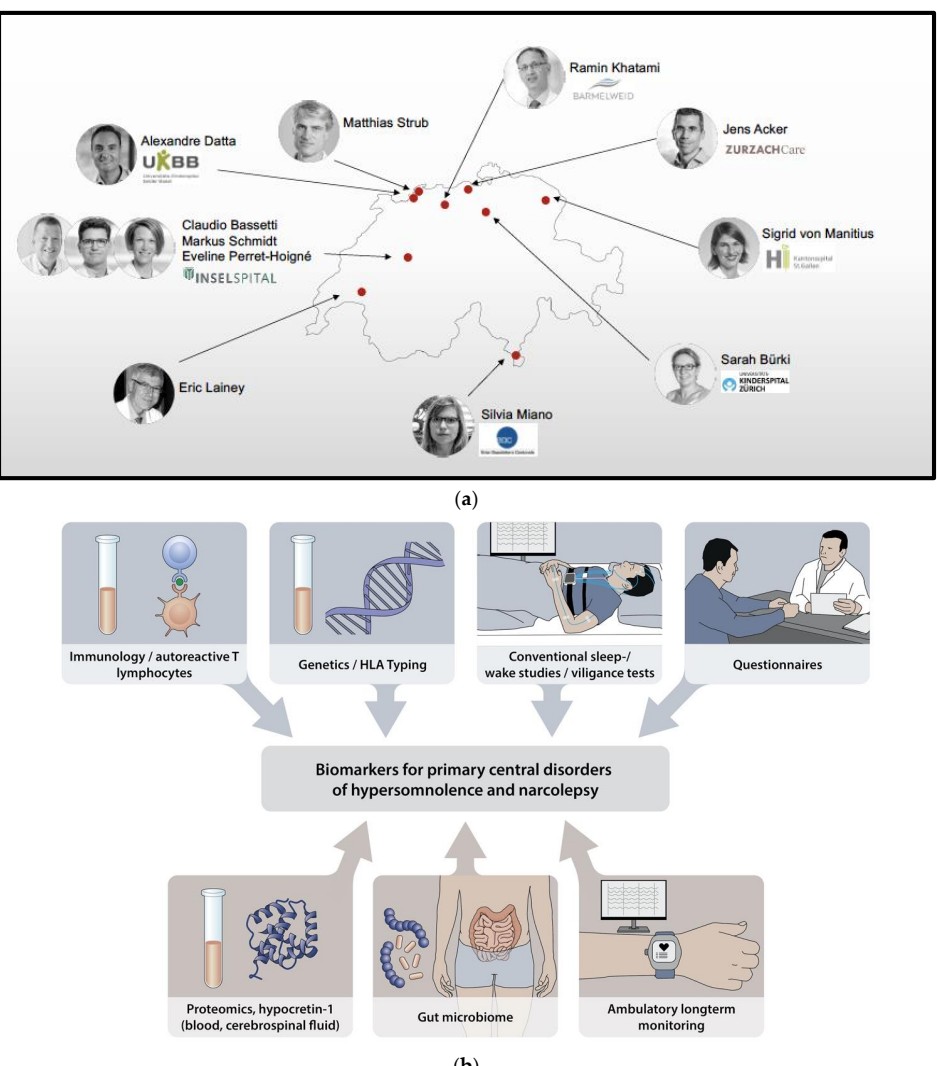

**Figure 2. SPHYNCS** (Swiss Primary Hypersomnolence and Narcolepsy Cohort Study) is a multicenter study supported by the SNF and conducted within SNaNe aiming to identify new biomarkers for narcolepsy and other primary central disorders of hypersomnolence. (**a**) The nine Swiss-certified sleep centers (four of them being pediatric) participating in SPHYNCS. (**b**) SPHYNCS aims to identify novel clinical, neurophysiological, digital, and "omics" biomarkers for primary central disorders of hypersomnolence [15]. A total of 170 patients have been enrolled into the study to date (September 2023; figure by A. Giger, Bern).

The webpage and newsletter (https://www.snane.ch/newsletter, accessed on 1 September 2023) provide regular information about the activities of SNaNe.

### 6. The Swiss Narcolepsy Network (SNaNe): Outlook and Perspectives

In the coming years, SNaNe will expand to become a Swiss competence network focused on the care of all patients with rare (neurological and non-neurological) sleep disorders in all language regions of Switzerland. Standardized operation diagnostic and therapeutic procedures will be developed and harmonized among the centers of this competence network, ensuring that patients are assigned as fast as possible to the specialists who carry the appropriate disease-specific knowledge. Additionally, SNaNe intends to establish and promote cooperation with the emerging Swiss centers for rare neurological and pediatric disorders in order to create low-threshold offers for affected persons or relatives, and to expand its functionality. Accordingly, the competence structure will be suitable for both patients with a known diagnosis of rare diseases, as well as patients with isolated or complex symptoms that do not fit existing disease criteria. Furthermore, the SNaNe is planning to implement an online communication platform. This point of contact is intended to enable and facilitate networking among various professional players, in addition to providing specialist medical advice, including the listing of scientifically tested offers. The exchange of social, medical, and psychological offers will be the primary focus herein.

**Author Contributions:** Conceptualization, R.K. and C.L.A.B.; writing—review and editing, S.M., E.W. (Elena Wenz) and E.W. (Esther Werth). All authors have read and agreed to the published version of the manuscript.

**Funding:** This research received no external funding.

**Institutional Review Board Statement:** Not applicable.

**Informed Consent Statement:** Not applicable.

**Data Availability Statement:** Not applicable.

**Conflicts of Interest:** The authors declare no conflict of interest.

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
