# Peer review of "The Swiss Narcolepsy Network (SNaNe)"

_ctn, doi:10.3390/ctn7040031_

Round 1

Reviewer 1 Report

Bassetti et al reported The Swiss Narcolepsy Network (SNaNe), it benefits patients and researchers. I hope it will share materials i.e., blood samples with other centers or labs.

Minors:

Line 41-49: The reasons should be listed from 1 to 5?

Should authors format Table 1?

Figure 1 is missing?

Line 73-74: haplotype DQB*0602, should be DQB1*06:02?

good except for a few typos.

Author Response

Many thanks for the suggestions of the reviewer which were all taken into account in the revised version of the manuscript.

Reviewer 2 Report

Thank you for the opportunity to review this manuscript. I think it can be published with some additions and corrections. Please, see my comments and suggestions below. 

Abstract: The numbers 1) etc are not written correctly, repetition. 

The prevalence of narcolepsy in the world is not clear and seems to vary, especially regarding the age strata investigated. Please elaborate this information and add recent references. One example from northern Europe revealing lower than expected numbers is by Gauffin et al but there may be others to add. (DOI: 10.1111/ane.13532).

Ad "hardly any research in this field": It is not true that research in the field of narcolepsy is lacking, please  modify by expanding the text and give recent examples.

It is also not true that pharmacotherapy often is successful. The drugs available are purely symptomatic, and many patients suffer from reduced vitality and workability, in addition cognitive problems which should be mentioned in the text, adequately referenced. Please modify this text.  (Ref. Witt et al. Evidence for cognitive resource imbalance in adolescents with narcolepsy. Brain Imaging and Behavior, 2018, Vol. 12, nr 2, s. 411-424)

Kleine Levin syndrome should be explained.

The mechanism of pathogenesis has been discussed, may be not solely a loss of hypocretine neurons but also a functional deficit. Elaborate.

About the European Narcolepsy Network - please give more details, for example the number of registered patients in the database. 

You may also give information about the impact of swine flu vaccination (Pandemrix) on the narcolepsy population. How was it documented by this network? In Switzerland? If negative, this is also of interest for the international reader.

The mentioning of inclusion of "all male and female" patients in several places seems strange. Natural to include both genders.

The map of the network includes photos and names of individual persons in charge. Reflect over the possibility to exclude these and just keep the map with the geographical information.

Fig. 2 b, legend: Error regarding fonts. Please correct.

There are some typos and misspellings in the reference list.

Author Response

We thank the reviewer for the detailed and very helpful comments/and suggestions, which we have all  taken into account in the revisions of the manuscript with two exceptions:

  • the purpose of this article in the presentation of the SNaNe, not of the single primary CDH. For this reason we suggest to avoid going into details (concerning the clinical presentations of primary CDH, etio-pathophysiology of narcolepsy, review of research progresses in the field) and prefer to refer recent review articles.
  • it is our wish to show not only the centers but also the leaders of the centers involved in SPHYNCS, this to give some visibility to a fantastic group of people in our country who made this joint effort possible

Unfortunately the SNaNe was not able to perform any studies on the effects of vaccinations.
